# Exosomes as Novel Delivery Systems for Application in Traditional Chinese Medicine

**DOI:** 10.3390/molecules27227789

**Published:** 2022-11-12

**Authors:** Qi Chen, Di Wu, Yi Wang, Zhong Chen

**Affiliations:** 1Key Laboratory of Neuropharmacology and Translational Medicine of Zhejiang Province, School of Pharmaceutical Sciences, Zhejiang Chinese Medical University, Hangzhou 310053, China; 2Department of Neurology, The Third Affiliated Hospital, Zhejiang Chinese Medical University, Hangzhou 310053, China

**Keywords:** exosomes, nanoparticle, drug delivery systems, traditional Chinese medicine

## Abstract

Exosomes, as gifts of nature derived from various cell types with a size range from ~40 to 160 nm in diameter, have gained attention recently. They are composed of a lipid membrane bilayer structure containing different constituents, such as surface ligands and receptors, from the parental cells. Originating from a variety of sources, exosomes have the ability to participate in a diverse range of biological processes, including the regulation of cellular communication. On account of their ideal native structure and characteristics, exosomes are taken into account as drug delivery systems (DDSs). They can provide profound effects on conveying therapeutic agents with great advantages, including specific targeting, high biocompatibility, and non-toxicity. Further, they can also be considered to ameliorate natural compounds, the main constituents of traditional Chinese medicine (TCM), which are usually ignored due to the complexity of their structures, poor stability, and unclear mechanisms of action. This review summarizes the classification of exosomes as well as the research progress on exosome-based DDSs for the treatment of different diseases in TCM. Furthermore, this review discusses the advantages and challenges faced by exosomes to contribute to their further investigation and application.

## 1. Introduction

Almost all prokaryotes and eukaryotes produce extracellular vesicles (20 nm to 2 μm) [1]. Exosomes belong to a class of extracellular vesicles secreted by different mother cells according to their sizes [2]. Typically, with a size range of ~40 to 160 nm, exosomes have a lipid bilayer structure storing nucleic acids, proteins, enzymes, and metabolites [3,4]. They perform key roles in cell-to-cell communication through the transfer of functional cargos to adjacent or distant cells [5]. Furthermore, exosomes can be easily transmitted into different areas of the body using routes such as the blood, urine, cerebrospinal fluid, and saliva [6].

Due to their capacities, they can be exploited as promising delivery systems to study various physiological and pathological processes [7,8,9]. Increasing evidence has also indicated that in the fields of cancer, atherosclerosis, Alzheimer’s disease, non-alcoholic fatty liver disease, and obesity, there has been much focus on exosomes [10,11,12,13,14].

As exosomes could provide drug delivery systems (DDS) with advantages that cannot be ignored, such as specific targeting, high biocompatibility, low toxicity, and editability, they could be a booster for those natural compounds to improve their stability and efficacy [15,16,17]. Natural compounds are known to be one of the main constituents of traditional Chinese medicine (TCM) [18]. Dozens of research studies have shown that biological activity from natural compounds is real [19]. However, many active ingredients are related to the mechanism, making them difficult for people to understand. Take curcumin as an example. James Inglese, who works at the National Center for Advancing Translational Sciences in Bethesda, Maryland, said “A lot of people doing this kind of work aren’t technically aware of all the issues that this thing can cause.” [20]. Thus, to make TCM better known, and to keep up with modern drug development, exosome-based DDSs will be a good tool. In this review, we will introduce the features of exosomes, illustrate the classification of exosomes, summarize the construction methods of exosome-based DDS, share some instances of their applications in TCM and discuss the challenges and opportunities. A systematic search was conducted mainly in Google Scholar, PubMed, Web of Science, and some official journal websites. Search terms were “exosome”, “TCM” and “drug delivery.”

## 2. Classification of Exosomes

Exosomes, microvesicles, and apoptotic bodies are the three main types of extracellular vesicles. Exosomes are nano-sized extracellular vesicles with cup-like shapes [21]. The other types are defined as being larger than exosomes. Microvesicles are 100–1000 nm, while apoptotic bodies are 500–2000 nm [22]. Meanwhile, the biogenesis processes of these three types are different [23]. Exosomes are released from the endosomal pathway after the formation of the multivesicular bodies (MVBs) and the fusion of MVBs with the plasma membrane (Figure 1). During this process, the invaginating membrane incorporates certain proteins. In the meantime, intraluminal vesicles engulf and enclose the cytosolic components. When it comes to this, it is necessary to refer to some hypotheses [24]. One is that an intricate protein machinery ESCRT is required for the formation of intraluminal vesicles in exosomal biogenesis. The full name of ESCRT is the endosomal sorting complex required for transport. It consists of four separate proteins that work cooperatively. Another hypothesis favors an alternative pathway in an ESCRT-independent manner, which seems to depend on raft-based microdomains. This hypothesis emphasizes the key role of exosomal lipids. In addition, within either the ESCRT-dependent or -independent mechanisms, there exist several specialized mechanisms, depending on the origin of the cell type, acting variously to ensure the specific sorting of bioactive molecules into exosomes.

### 2.1. Naive Exosomes

Naive exosomes can be divided roughly into two categories: eukaryotic exosomes and prokaryotic exosomes. All animal cells and plant cells are eukaryotic cells while bacteria and cyanobacterial cells are prokaryotic. A summary of the naive exosome source is given in Figure 2.

Animal exosomes are released from normal cells and diseased cells., including fibroblasts, immune system cells (T- cells, B-cells, dendritic cells), adipocytes, and tumor cells and isolated from all normal and diseased body fluids such as blood, urine, breast milk, amniotic fluids, and bronchial alveolar lavage. All these cells have been proven to play important roles in normal or abnormal physiological processes. For blood cells, reticulocytes, red cells, white cells, and platelets are the most commonly studied blood cells. Qi H. et al. demonstrated blood is a good source of exosomes [25]. During erythrocytes’ maturation from reticulocytes, there are at least 200 μg (10^14^) exosomes produced per day [26]. Blood exosomes are safe and contain various membrane proteins, including transferrin (Tf) and CD47 receptors, and thus can escape from the immune system [27]. Qu M. et al. reported that blood exosomes had a good natural brain-targeting ability [28]. The mechanism involves the transferrin-transferrin receptor interaction. For immune cells, antigen-presenting cells, myeloid-derived suppressor cells, T regulatory cells, T lymphocytes, and NK cells are the main categories [29]. Here we take macrophages that belong to antigen-presenting cells as examples. Yuan D. et al. examined the possibility of macrophage exosomes penetrating the blood–brain barrier (BBB) in preclinical animal models [30]. Zhu J. et al. used crosstalk between macrophage exosomes and vascular smooth muscle cells (VSMCs) to regulate atherosclerosis progression [31]. For cancer cells, there are countless categories such as colon cancer cells, melanoma cells, breast cancer cells, etc. Cancer cell exosomes transfer not only between cancer cells but also between stromal cells and cancer cells [32]. It is worth mentioning that cancer cell exosomes as antigens could provide a specific immune response and homologous targeting behavior. Typically, cancer cell exosomes cross-present the antigen to dendritic cells (DCs) and then activate T cells, or directly activate NK cells or macrophages [33,34,35]. Zhou W. et al. reported that a pancreatic ductal adenocarcinoma (PDAC) exosome-based system could induce effective innate and adaptive anti-PDAC immunity through improved DC maturation and cytotoxic T lymphocyte infiltration [36]. Ji Y. et al. found that colon cancer cell-derived exosomes inhibit the proliferation and migration of homologous colon cancer cells [37].

Exosomes which are secreted by plants, such as from ginger [38], grapefruit [39], grape [40], carrot [41], broccoli [42], coconut [43], and apple [44], have recently been attracting attention. These exosomes are readily available without toxic effects and are easily applied via the oral route in animals [45]. They offer obvious advantages stemming from the biochemicals of the original plants. For example, ginger exosomes contain many active components including 6-gingerol and 6-shogaol, which have the potential for anti-oxidative, anti-inflammatory, and anti-cancer activities [46]. Zhang M. et al. illustrated ginger exosomes can be produced on a large scale [47]. Sundaram K. et al. demonstrated that ginger exosomes interact with host hepatocytes and then boost the anti-inflammatory efficiency [48]. Grape exosomes are enriched with PE (26.1%) and phosphatidic acids (PA) (53.2%). Wang Q. et al. fabricated grape exosome-based nanoparticles and proved that they are less toxic than nanoparticles made of synthetic lipids [40].

Bacteria exosomes are essential for the survival and development of bacteria. Exosomes released from Gram-negative bacteria are also known as outer membrane vesicles (OMVs) [49]. Previously, it was believed that the exosomes were specific to Gram-negative bacteria because of the presence of a complex cell wall with a peptidoglycan (PG) layer, which may restrict the release of exosomes from Gram-positive bacteria [50]. Recently, compelling evidence has shown not only that Gram-negative bacteria produce exosomes, but also that Gram-positive bacteria could release them [51]. Bacteria exosomes have been examined extensively concerning their application in vaccine development [52], infection control [53], cancer treatment [54], and bioimaging [55].

To sum up, protein, mRNA, miRNA, DNA, lipids, and metabolites constitute the biological composition of naive exosomes in the inner cavity [56]. They vary depending on the source and the original isolation or enrichment techniques. Eukaryotic exosomes are secreted by cells that are surrounded by a single membrane. Typically, they have soluble factors in the inner area, such as hormones, growth factors, and cytokines that can act on the source cell itself. Prokaryotic exosomes, focused on the pathogenic bacteria, are secreted by Gram-negative bacteria that possess two phospholipid membranes and Gram-positive bacteria that are surrounded only by a single membrane, which contains toxins and virulence factors such as the lipopolysaccharides (LPS) [57] (Figure 3).

### 2.2. Modified Exosomes

Irrespective of the parent cell, exosomes have some common features, including the expression of certain tetraspanins (CD9, CD63, and CD81), heat shock proteins (Hsp 60, Hsp 70, and Hsp 90), biogenesis-related proteins (Alix and TSG 101) and so on [58]. We can also check the collected data in Vesiclepedia (http://www.microvesicles.org, accessed on 1 October 2022), Exocarta (http://www.exocarta.org, accessed on 1 October 2022), and from a Bioinformatics lab in China (http://bioinfo.life.hust.edu.cn, accessed on 1 October 2022), to know their distinguishing characteristics [59]. Although there are thousands of characteristics that exosomes could provide by nature, their practical applications are still limited in some cases. Therefore, to create exosomes with more of the desirable features, genetic and chemical engineering have been developed to modify exosomes (Figure 4).

Genetic engineering allows the designed gene sequence of a guiding protein or polypeptide into exosomes. This method is good for surface display, but the decorative motif is limited. In brief, designed ligands are first fused with the expressed transmembrane proteins. After that, the plasmids encoding the fusion proteins are prepared. Finally, donor cells transfected with plasmids secrete genetically engineered exosomes containing the designed ligands. Here are some examples of genetically engineered exosomes. The tetraspanin superfamily CD63/CD9/CD81 [60], lysosome-associated membrane protein (LAMP) family LAMP-2B [61], and the transmembrane protein platelet-derived growth factor receptor (PDGFR) are exosomal surface proteins commonly used to display designed ligands [62]. The tetraspanin superfamily CD63/CD9/CD81 has two extracellular loops that can be engineered [63]. It is reported that combining CD63 with ovalbumin (OVA) antigen could produce OVA exosomes to fabricate DNA vaccines with improved immunogenicity [64]. The N-terminus of LAMP-2B can be appended with engineered sequences [65]. LAMP-2B fused with integrin-specific iRGD peptide (CRGDKGPDC), could produce iRGD exosomes which target integrin-positive breast cancer cells [66]. Between the N-terminal signal peptide and the transmembrane domain of PDGFR, genetically engineered motifs can be inserted. PDGFR transmembrane region fused with GE11 (YHWYGYTPQNVI) could secrete GE11 exosomes which show a high affinity for epidermal growth factor receptor (EGFR)-overexpressing cancer cells [67].

Chemical engineering uses conjugation reactions or lipid assembly to fuse natural or synthetic ligands. However, the complexity of the exosome surface may hamper the processes; furthermore, some reactions lack site specificity and even increase the toxicity of exosomes. An ideal used method for conjugation to exosomes is bioorthogonal reaction using non-copper catalyzed click chemistry. Reagents of click chemistry include the dibenzocyclooctyne (DBCO)-based reagents [68], tetrazine-based reagents [69], azide-based reagents [70], alkyne-based reagents [71], trans-cyclooctene (TCO)-based reagents [72], cyclopropene-based reagents [73], and cleavage linkers [74], etc. In one study, dibenzocyclooctyne-modified signal-regulating protein α (SIRPα) and CD47 antibodies were clicked with azide-modified M1 macrophage exosomes through pH-sensitive linkers [75]. This reaction by click chemistry relies on amine groups on exosomes converse to alkynes. Lipid assembly means the insertion of amphipathic external objective lipids into the original lipid bilayer of subjective exosomes. Objective lipids like DSPE-PEG [76], cholesterol [77], and related derivatives are widely studied [78]. For instance, Cui Y. et al. prepared a bone-targeting-peptide-linked DSPE-PEG-Mal and developed a bone-targeted engineered mesenchymal stem cells (MSCs) exosome platform [79]. Leaving aside the use of artificial lipids, some natural lipids can also be used as objects to create hybrid exosomes. Hu S. et al. functionalized stem cell exosomes with platelet membranes to enhance the binding and accumulation of exosomes in injured tissues [80]. This DDS used platelet membranes to modify stem cell exosomes. Platelet membranes are lipid bilayers having membrane integrin receptors, such as GPVI, participating in the natural “injury finding”. Hence, platelet membrane-hybrid exosomes can target vascular injury under myocardial infarction. Compared to non-modified exosomes, hybrid exosomes notably enhanced the cellular uptake in endothelial cells and car-diomyocytes, but not in macrophages. Furthermore, platelet membranes improve the therapeutic capacity of stem cell exosomes. Liver accumulation is undesirable and should be avoided, as this DDS is not intended for liver diseases. However, Hu S. et al. found a steady concentration of platelet membrane-hybrid exosomes in the liver. Their future studies will try to fine-tune the exosomes to reduce off-target loss into the liver.

## 3. Construction of Exosomes as Drug Delivery Systems

An ideal DDS tends to have the following features: (1) excellent biocompatibility with no toxicity, (2) high stability in the blood and other bodily fluids, evading clearance by the innate immune system including both humoral and cellular immune systems, (3) high efficiency in targeting the destination, (4) great uptake and release behavior toward desired cells, (5) great biodegradability [81].

Exosomes, as one of the rising biomimetic materials to construct DDSs, have been explored in recent years. Many scientists have been interested in the development of exosome-based DDSs for improving the uptake. For example, Piffoux M. et al. pointed out that a liposome–exosome hybrid DDS structure could improve cellular uptake of anti-cancer photosensitizer in CT26 colon cancer cells from approximately 20% to 70% [82]. Mentkowski K. et al. demonstrated that cardiomyocytes exhibited increased uptake of cardiomyocyte-specific peptide-modified exosomes by two-fold when compared with non-targeted exosomes [83]. Likewise, Ou Y. et al. investigated cellular uptake mechanisms of exosomes showing an 8-to-40-fold higher cellular internalization than liposomes, mainly through receptor-mediated processes (i.e., clathrin-and caveolae-mediated endocytosis) [84]. The construction of exosomes as drug delivery systems concerning their pros and cons are discussed as follows.

### 3.1. Exosomes Isolation and Modification

The construction of exosomes as DDSs starts from the isolation of exosomes from the mother cells. Physical and biological characteristics of exosomes are utilized to separate them, mainly including density or size and impurities. Examples of exosome isolation methods are summarized in Table 1.

Ultracentrifugation is the most widely used technique to separate exosomes based on particle size and density. Usually, there are two kinds of ultracentrifugation, differential ultracentrifugation and density gradient ultracentrifugation. The procedure is carried out step by step. It is recognized as an easy operation that induces little contamination [85,86]. Firstly, remove the large cell fragments at a low speed. Then, remove the small fragments at an increased speed. As the centrifugation speed increases, a refrigerated floor-standing ultracentrifuge instrument is required. Because of this, the operation time is quite long, and it is necessary to transfer the exosome-containing solutions multiple times. The yield is also affected by the frequent transfer. The difference between differential ultracentrifugation and density gradient ultracentrifugation is that the latter uses density gradient solvents from low to high strengths to ensure exosomes can migrate to the equilibrium density through centrifugation [87,88,89].

Ultrafiltration is the procedure that requires ultrafiltration tubes bearing a semi-permeable membrane with a certain pore size. Different pore sizes correspond to different molecular weights. Exosomes with smaller molecular weights could pass the corresponding pore sizes. Those with larger molecular weights are blocked. This is a concise method without affecting the biological activity of exosomes [90,91]. However, it still needs centrifugation.

Size exclusion chromatography consists of the stationary phase and the mobile phase. The stationary phase uses small porous polymer beads. When the exosome solution passes through the stationary phase, smaller exosomes can enter the beads while the larger particles can only follow the mobile phase to be washed out [92]. The size exclusion chromatography method can produce exosomes with good homogeneity. Meanwhile, it is a less time-consuming, more convenient, and more automatic method when compared to the above centrifuge method.

Immunoaffinity capture relied on the specific affinity of antigens and antibodies. Some surface molecules on exosomes could be known as specific binding sites. Due to the high specificity, this simple operation takes only a few minutes to complete the separation. Furthermore, it does not affect the integrity of exosome morphology. However, this process needs strict environmental conditions because pH and salt concentration can influence biological activity [93,94,95]. For downstream experiments, it is not conducive and even difficult to use this technique widely.

Membrane-based separation is a new and powerful isolation technology that is implemented by negatively charged phosphate groups in lipid bilayer structures of exosomes [96,97]. Some metal oxides (e.g., TiO_2_) and positively charged molecules (e.g., protamine) can easily bind to the phosphate groups. Because exosomes are rich in phosphate groups, this strategy for isolating exosomes has obvious advantages in high yield and speed. However, for some similar structures of exosomes in the microenvironment in which exosomes lie, membrane-based separation may result in other impurities.

Polyethylene glycol precipitation uses the polymer to reduce the dissolution of exosomes in the aqueous solution. Exosomes could be precipitated and aggregated in large quantities. The aggregates could be separated through low-speed centrifugation. However, co-precipitation is indiscriminate for all proteins. The soluble non-exosomal proteins, immunoglobulins, viral particles, immune complexes, and other contaminants may also be found in the final pellet from polyethylene glycol precipitation [98].

Microfluidic approach is a high-throughput technique that uses devices to distinguish different nanoparticles based on various principles such as size, density, charges, and immunoaffinity [99,100]. It is a new technology that can reduce the operation time for exosome separation and cost less in samples and reagents. However, because of the high cost, the microfluidic approach is not yet considered a standardized method [101].

From what have discussed above, the choice of which isolation method to use in which case should depend on the downstream application together with the amounts of starting materials [102]. Not one EV isolation approach is suitable for all forms of exosome study [103]. The following are some recommendations from the authors. If the application needs a high quantity of exosomes, the precipitation method is the best. If the size range matters, ultracentrifugation, size exclusion, chromatography, and immunoaffinity capture can be chosen for their yield of a narrow size range for exosomes. If starting materials are limited, methods that cause small consumption of samples need to be picked, such as ultrafiltration and the microfluidic approach. In addition, combined exosome isolation methods can be considered to improve the exosome extraction [102].

After successfully collecting exosomes, it is sometimes still necessary to modify the exosomes to transform them into more capable vesicles. Exosome modification strategies can be found in Section 2.2.

### 3.2. Drug Loading Methods

There are two main methods to load drugs into exosomes, exogenous drug loading and endogenous drug loading [104]. The details of these two are elucidated below.

The steps of exogenous drug loading methods and endogenous drug loading are reversed in relation to each other. For exogenous drug loading, exosomes are first collected and then loaded with drugs. For endogenous drug loading, drugs are first loaded into cells and then the cells secrete exosomes with drugs. The loading methods include incubation, electroporation, sonication, extrusion, freeze-thaw, acoustofluidics, and hypotonic dialysis.

Incubation can be used in exogenous and endogenous drug loading. Those hydrophobic or small-molecule drugs can be directly incubated with cells or exosomes. Hydrophilic compounds cannot passively pass through lipid vesicles. Hence, electroporation, sonication, extrusion, and freeze-thaw methods can create pores on the exosomes to allow the hydrophilic drugs to enter [105]. Table 2 summarizes the drug-loading methods for exosomes.

Incubation is the most straightforward loading method. The operation is simple, but its efficacy is limited. Saari H. and Garofalo M. et al. dissolved PTX and incubated it with 1 × 10^8^–5 × 10^9^ exosomes from prostate cancer cells for 1 h at 22 °C. After incubation, they collected samples to centrifuge at 170,000× *g* for 2 h to pellet the drug-loaded exosomes. This loading strategy yielded an average loading efficiency of 9.2% [106,107]. Li Y. et al. fabricated DOX-loaded exosomes. First, Dox HCl was dissolved in the phosphate-buffered solution (PBS) and desalinized with trimethylamine. Then, 20 μg Dox was mixed with 200 μg exosomes for 5 min and dialyzed overnight. The Dox encapsulation efficiency and loading capacity were about 9.06% and 2.60%, respectively [108]. Wu T. et al. exposed BMDCs to ultraviolet irradiation for 1 h. DOX was then added to the culture medium. The supernatant was centrifuged to collect DOX-loaded exosomes from BMDCs after 16 h. The loading amount of DOX by exosomes was correlated with the initial DOX concentration [109].

Electroporation is the most common endogenous strategy for transducing desired nucleic acids, proteins, and peptides into exosomes, and drugs were loaded afterward. This offers potential advantages in loading efficiency and molecular stability. However, the toxicity associated with the transfection reagent may result in undesirable changes in exosome cargo and bioactivity. Tian Y. et al. engineered the imDCs to express a well-characterized exosomal membrane protein (Lamp2b) fused to αv integrin-specific iRGD peptide (CRGDKGPDC) and collected 100 mg of purified exosomes after transfection by centrifugation [66]. To load the transfected exosomes with DOX, they mixed purified exosomes and DOX in 200 mL of electroporation buffer at 37 °C for 30 min. The electroporation was set under 350 V and 150 mF in 0.4 cm electroporation cuvettes via a Gene Pulser II Electroporator (Bio-Rad, Hercules, CA, USA). They detected the intrinsic fluorescence of Dox to quantify the loading mass, but the specific loading amount was not shown in their publication. Hood J. et al. loaded B16 melanoma exosomes with superparamagnetic iron oxide nanoparticles by electroporation [110]. They used a unique trehalose pulse media (TPM) which could minimize the aggregation to suspend superparamagnetic iron oxide and B16 melanoma exosomes. Then, a BTX Harvard Apparatus ECM 399 system (Holliston, MA, USA) was used for electroporation. Iron loading of exosomes shifted peak melanoma exosome density from 1.14 g/mL to 1.180 g/mL by electroporation. Nakase I. and Futaki S. loaded fluorescently labeled dextran in exosomes from HeLa cervical cancer cells through electroporation [111]. The settings which poring pulse (twice pulse, 5 ms and transfer pulse (five pulses, 20V, 50 ms)) were completed using a super electroporator NEPA21 TypeII (NEPA genes, Tokyo, Japan) in a 1 cm electroporation cuvette at room temperature. The final concentration of fluorescently labeled dextran was 150 ng/mL in 20 μg/mL encapsulated exosomes.

Sonication is a physical exogenous procedure that relies on an extra mechanical shear force to weaken the structure of exosomes, promoting the loading of drugs. It shows a higher loading capacity than other strategies, but it may damage the integrity of exosomes. Lee E. S. et al. entrapped DOX into mouse macrophage RAW 264.7-based exosomes using the sonication method [112]. HCl-detached DOX was added to PBS-contained exosomes. The mixture was sonicated using a tip sonicator with 20% amplitude and 6 cycles of 30 s on/150 s off. The DOX loading content in the exosomes was in the range of 7.3–9.9 wt.%. Kim M. et al. mixed purified exosomes (~10^11^ exosomes) with drug PTX and DOX and sonicated them for six cycles of 30 s on/off [113]. The total operation time is 3 min with a 2 min cooling period. After sonication, they obtained PTX/DOX-loaded exosomes which had good stability for over a month, taking the size, the zeta potential, and the quantity of drug loading into account. The obtained loading capacity was 28.29 ± 1.38%. Lamichhane T. et al. explored the use of sonication as a method to load small RNA in exosomes [114]. They optimized the sonication parameter and found sonication can increase the loading amounts to 2.96% when compared to electroporation.

Extrusion is also a physical exogenous procedure that extrudes exosomes and drug mixtures in an extruder to cause the recombination of exosomal membranes. Repeated extrusion provides a homogenous blend of exosomes. However, the process is resource-consuming because lots of solutions containing exosomes may be lost through the operation. Chen Q. et al. presented functionalized exosomes based on attenuated Salmonella and further coextruded them with tegafur-loaded polymeric micelles to improve the synergistic therapeutic effect between bacterial exosomes and drugs [115]. They used 220 nm polycarbonate membranes, and the extrusion process was repeated seven times to ensure that the drug micelles could be completely covered by bacterial exosomes. The drug loading efficiency was 7.13%. Cheng G. et al. developed a biomimetic nanoparticle platform based on exosomes secreted by MDA-MB-231 tumor cells and metal−organic frameworks (MOFs)-protein drugs for systemic and intracellular delivery of proteins [116]. Guest proteins were caged in the MOFs with up to ∼94% high efficiency and then act as the cores self-assemble into exosomes by ultrasonication and repeated extrusion.

Freeze-thaw methods are exogenous procedures, commonly used for loading drugs into exosomes by temporarily forming ice crystals to disrupt the integrity of the exosomes. This method is simple and effective for loading various cargoes. However, repeated free-thaw cycles may cause undesired degeneration and aggregation of exosomes. Cheng L. et al. fabricated hybrid therapeutic nanovesicles, fusing gene-engineered exosomes with drug-loaded thermosensitive liposomes by using the freeze-thaw method [117]. Gene-engineered exosomes were collected from CD47-transfected CT26 cells. Indocyanine green (ICG) and Imiquimod (R837) were the model drugs to encapsulate into liposomes. The drug-loaded thermosensitive liposomes and the gene-engineered exosomes were mixed at a ratio of 1:1; then, the mixture was frozen for 15 min at −80 °C and incubated for 15 min at 37 °C for 3 cycles in total. The final hybrid therapeutic nanovesicles presented 66.9% CD47 positive.

Acoustofluidics have great potential for nanoparticle assembly and controllable drug release in both exogenous loading and endogenous loading. This facilitates exosome encapsulation during drug loading, enabling the recruitment of exosomes. The main bottlenecks are cost, scalability, validation, sample pretreatments, and standardization. Wang Z. et al. demonstrated their drug loading method based on acoustofluidics, which can within a single-step process simultaneously concentrate drug molecules with silica nanoparticles and encapsulate the particles within exosomes [118]. When compared to the passive incubation method, which only has 0.08% loading efficiency, acoustofluidic drug loading achieved a higher loading efficiency of ~30% within 2 min.

Hypotonic dialysis relies on a transmembrane pH gradient to encapsulate weakly basic or negatively charged small-molecule drugs effectively. It could be defined as both exogenous drug loading and endogenous drug loading. However, some high molecular weight proteins in the exosomes may degrade using the dialysis steps. Wei H. et al. prepared exosome-loaded doxorubicin (Exo-Dox) by mixing both of them, desalinizing them with triethylamine, and dialyzing them with PBS overnight [119]. The Dox encapsulation efficiency was about 12%. Jeyaram A. et al. exposed exosomes to dehydration by ethanol for miRNA loading [120]. They optimized the parameters for nucleic acid loading and observed that room temperature (22 °C) and 2 h incubation times were the best. The highest loading efficacy they obtained was 6.5%.

## 4. The Application in Chinese Medicine

Due to the rapid development of nanotechnology, the systematic control of drug release and delivery could be realized by nano drug-delivery systems. Evidence to date suggests that the stability of the exosomes as drug delivery systems over time and upon temperature is good [121]. They extend the drug’s half-life and increase drug release stability by encapsulating the drug inside the lipid bilayer membrane structure. It was reported, under freezing conditions (−20 °C and −80 °C), that no significant change happened in the inner content [122]. For up to two years, exosomes remain active as drug delivery systems [123].

Using TCM-encapsulated and TCM-primed exosome-based drug delivery systems is an interdisciplinary area of research in science, engineering, and medicine. Natural compounds extracted from TCM, including paclitaxel (PTX), camptothecin, curcumin, tanshinone, vincristine and catalpol, etc., involving multiple targets and multiple signaling pathways, have become popular candidates (Figure 5) [124]. However, due to their complexity of structures, poor stability, and unclear mechanisms of action, natural compounds have usually been ignored before and are hardly used in clinical settings. In addition, challenges with respect to the stability and storage of the exosome-based DDSs still persist because this mode may be limited by cost. This section introduces the related research progress of exosome-based drug delivery systems to solve the problems that have limited the applications of Chinese medicine.

### 4.1. Oncology

Cancer is the leading cause of death in every country of the world [125]. Efforts should be made to control the increasing burden of cancer. Surgical treatment, chemotherapy, radiotherapy, and immunotherapy are the main types of treatment methods for cancer [126]. However, using traditional single treatment methods cannot meet all the challenges of recurring and metastasizing tumors. In the case of chemotherapy, multidrug resistance (MDR) has become the norm [127]. Therefore, combination therapies have become an increasingly crucial part of the realm of oncology. TCM-combination therapy, which thinks of cancer as a systematic disease associated with the state of the whole body, is a new strategy, especially the delivery of TCM, which exhibits promising antitumor potential in the treatment of a variety of tumors [128].

One of the classic TCM is Paclitaxel (PTX). PTX was isolated from Taxus brevifolia [129]. Its anticancer activity was first confirmed in 1971. The anti-tumor mechanisms are still under investigation. One has proposed that PTX can promote tubulin polymerization and stabilize microtubules [130,131]. Effects that PTX could induce in experimental models, such as inhibition of angiogenesis, induction of cytokines such as GM-CSF or IL-2, induction of apoptosis, and interaction with the immune system, have aroused much interest in treating cancer [132]. Pascucci L. et al. first demonstrated that mesenchymal stromal cells (MSCs) can package and deliver PTX through their exosomes [133]. They investigated the roles of exosomes from mesenchymal stromal cells (MSCs), which can home in on the tumor mass, in the releasing mechanism of PTX. The human cell line CFPAC-1 (human pancreatic adenocarcinoma that is very malignant and insensitive to chemotherapy) was used as the tumor model and the murine SR4987 line was used as the MSC model. The study compared the inhibitory activity of SR4987 exosomes carrying PTX (SR4987PTX-CM) and pure PTX on CFPAC-1 and found an equivalent drug concentration released by SR4987PTX-CM). Furthermore, they have shown that SR4987PTX-CM still retains a significant anti-tumor effect and pharmacological activity of PTX. Wang P. et al. isolated exosomes from M1 and M2 phenotype macrophages (M1-Exos and M2-Exos) to prepare the PTX delivery system (PTX-M1-Exos and PTX-M2-Exos) for breast cancer [134]. From their results, M1-Exos could activate the NF-κB pathway, create a pro-inflammatory environment, up-regulate the expression of cytokines iNOS, IL-6, and IL-12, and down-regulate the expression levels of cytokines IL-4 and IL-10, which enhanced the anti-tumor activity of PTX. However, the PTX-M2-Exos group showed opposite results, with no obvious improvement in anti-tumor activity when compared to that of the PTX group.

Another classic ingredient is Curcumin (Cur), a polyphenolic compound that originates from the tropical plant Curcumin longa’s root [135]. Studies have shown Cur could suppress tumor cell proliferation, especially for tumor stem cells [136,137]. The anticancer mechanisms of Cur are varied [138,139,140]. Compared to other conventional drugs for chemotherapy, Cur showed fewer adverse effects [141]. Jia G. et al. fabricated RGERPPR peptide (RGE)-modified RAW264.7 exosomes and loaded them with SPIONs and Cur, named RGE-Exo-SPION/Cur, using electroporation and click chemistry [142]. RAW264.7 exosomes themselves can easily cross the blood–brain barrier (BBB) [143]. The RGE peptide is a specific ligand of a transmembrane glycoprotein Neuropilin-1 (NRP-1), which is overexpressed in glioma cells and the tumor vascular endothelium [144]. SPIONs could work as image agents for gliomas [145]. This strategy proved the RGE-Exo-SPION/Cur could smoothly cross the BBB, accurately recognize gliomas, steadily improve the anti-tumor effect of Cur, and easily overcome the deficiencies of Cur and SPIONs. Zhang H. G. reported that exosomes from monocyte-derived myeloid cells can deliver curcumin to activated myeloid cells, playing important roles in cancers [146]. They thought that exosomes contribute more than the curcumin carriers to solve their major unstable barrier for clinical use. Exosomes also take part in the regulation of the immune response and the progression of tumors.

### 4.2. Cardiovascular Diseases

Cardiovascular disease (CVD) accounts for many deaths all over the world. Here is a 2021 summary of cardiovascular health and diseases in China, saying two out of every five deaths were due to CVD. Stroke, coronary heart disease, atrial fibrillation, rheumatic heart disease, congenital heart disease, lower extremity artery disease and hypertension are common cardiovascular diseases placing a high burden on human health [147]. Moreover, people with mental illness are under more severe conditions, with twice the cardiovascular mortality rate as the general population [148]. Limitations exist in modern clinical treatment. Hence, TCM can be used as a complement.

Buyang Huanwu decoction (BYHWD), composed of seven kinds of Chinese herbs, Huangqi (Radix astragali seu hedysari), Danggui (Radix angelica sinensis), Chi Shao (Radix paeoniae rubra), Chuanxiong (Rhizoma ligustici chuanxiong), Honghua (Flos carthami), Taoren (Semen persicae), and Dilong (Pheretima), is a herbal prescription for treating stroke [149]. Run X. et al. used BYHWD to treat mesenchymal stem cells (MSCs), gathered exosomes from treated MSCs, and studied the mechanism through which exosomes from BYHWD-treated MSCs have the capacity to protect mice against stroke [150]. From their results, the BYHWD exposure augmented MSCs’ behaviors, not only in vascular density, but also in VEGF and Ki-67 expression.

Suxiao Jiuxin pill (SJP), which contains two principal components, tetramethylpyrazine (TMP) and borneol (BOR), is a popular TCM for treating acute coronary syndrome [151,152]. Tang Y. L. et al. investigated exosomes from SJP-treated cardiac mesenchymal stem cells (SJP-Exos), TMP-treated cardiac mesenchymal stem cells (TMP-Exos), and BOR-treated cardiac mesenchymal stem cells (BOR-Exos), respectively [153]. In this study, they observed that TMP-Exos and BOR-Exos were similar to SJP-Exos and can both increase histone 3 lysine 27 trimethylation (H3K27me3) levels in recipient cardiomyocytes. H3K27me3 reflects the fates of stem cells and cardiomyocytes, such as regeneration, cardiac reprogramming, cell survival, and proliferation [154,155]. However, for demethylases, treatment with SJP-Exo selectively suppressed their expression. Groups of SJP-Exos showed the best effects in cardiomyocyte proliferation promoting, suggesting that the multi-component TCM compound has advantages over each component.

### 4.3. Others

Inflammatory arthritis is a chronic disease with a complex pathophysiological process [156,157,158]. If not treated in time, it could cause pain in the joint and even worse loss of joint function [159]. Rheumatoid arthritis (RA), represented by synovial hyperplasia and progressive joint destruction, and osteoarthritis (OA), characterized by cartilage loss and progressive joint degeneration, are both forms of arthritis driven by multiple inflammatory and metabolic factors [160]. Although glucocorticoids and nonsteroidal anti-inflammatory agents have been exploited in arthritis treatment, the efficacy is still unsatisfactory in clinical settings [161]. In addition, glucocorticoids and nonsteroidal anti-inflammatory agents could induce side effects such as teratogenicity and hepatotoxicity [162,163,164]. It is more desirable to find natural derivatives to solve the problems of strong side effects. Berberine (BBR) is a naturally occurring isoquinoline alkaloid, originating from several TCMs such as Coptis Chinensis, and has been proven to inhibit inflammation by suppressing dendritic cells’ activation and inflammatory proliferation of macrophages [165,166]. Ma Q. et al. delivered BBR with platelet-derived exosomes (PEVs) to the inflammatory environment in RA joints [167]. BBR-PEVs were traced in affected joints with high accumulation. After the BBR-PEVs treatment, the mobility of arthritis mice improved. When compared to the free BBR treatment group, at the same dose, the BBR-PEVs treatment group significantly reduced the infiltration of inflammatory cells and regulated the phenotype of dendritic cells and macrophages. This DDS ensured the efficacy of RA treatment with exquisite biocompatibility.

Spinal cord injury (SCI) is a serious neurologic insult which differs from peripheral nerve injury, resulting in permanent damage and motor, sensory, and autonomic dysfunction [168]. There are roughly two stages of SCI: primary injury and secondary injury [169]. As people’s understanding of neuroscience deepens, more interventions are geared at the secondary injury, which mainly refers to the inflammatory reaction. Secondary injury is expected to become the hope for neuronal regeneration and functional restoration [170]. Gao Z. et al. reduced inflammation during secondary injury using exosomes from M2-type primary macrophages to load berberine (Exos-Ber) [171]. They proved that the properties of Exos-Ber were stable and exosomes extracted from M2 macrophages can sustain their surface proteins. This drug delivery system has the ability to penetrate the blood–brain barrier, target macrophages/microglia in the brain and injury site and regulate the polarization of macrophages from proinflammatory M1 phenotype to anti-inflammatory M2 phenotype.

Inflammatory bowel disease (IBD) is a chronic intestinal disease, which places a heavy health burden on people worldwide [172]. The IBD condition includes two main forms—ulcerative colitis and Crohn’s disease [173]. Ulcerative colitis damages the mucosal layer of the colon while Crohn’s disease damages the mouth, anus, and entire layers of the intestine [174]. These two forms have similar symptoms to digestive system disorders [175]. Nowadays, oral chemical drugs such as 5-aminosalicylic acid derivatives, corticosteroids, and immunosuppressants are the classical chemotherapies for IBD [176]. However, the therapeutic outcome is still unsatisfactory. Patients with IBD often need to take the drugs for a lifetime [177]. It is necessary to find a promising strategy to improve long-time biosafety. Zu M. et al. isolated and purified exosomes from tea leaves to treat IBD [178]. Extraction from tea leaves ensures that the exosomes can be produced on large scale, which means this strategy is easy and cheap. They analyzed the major contents of these natural exosomes from lipid and protein composition profiles. The presence of galactose facilitated the internalization of exosomes by macrophages. Furthermore, abundant biological functional molecules including polyphenols and flavones have presented antioxidant activity in exosomes from tea leaves. By reducing ROS levels and the expression of pro-inflammatory cytokines, such exosomes are effective in alleviating the symptoms of colitis without apparent side effects. Their results illustrated that tea exosomes are an economically feasible platform for the prevention and treatment of IBD through the oral route.

## 5. Conclusions

### 5.1. Conclusive Remarks

Exosomes, key counterparts in cell-to-cell communication, have become a very popular topic. Meanwhile, recent decades have witnessed substantial improvements in TCM. It is worth discussing how to clearly understand the link between the exosomes and TCM, and how to combine them together to contribute more meaningful work. Thus, in this review, we have classified exosomes roughly into naïve exosomes and modified exosomes as well as referring to the construction of exosome-based DDSs in two parts, including exosome isolation and modification, and drug loading methods for the treatment of different diseases in TCM.

We can definitely consider exosomes as gifts of nature. Naive exosomes have a wide range of sources as all eukaryotic cells and prokaryotic cells could secrete them. Furthermore, modified exosomes contribute more classes to the field of exosomes. Because they mediate cell-to-cell communication, which is under the holistic theme of TCM, exosomes have broad prospects in TCM. In addition to exogenous exosomes, exosomes secreted by plants, which are the central element of TCM, also play important roles in the treatment of many diseases. They can also easily carry substances and target orientation, showing high stability and strong biocompatibility in vivo. Therefore, we think TCM-encapsulated and TCM-primed exosome-based DDSs are both equally important.

### 5.2. Future Recommendations

Although there are various relevant literature reports about exosomes as novel delivery systems and the current methods to characterize exosomes and exosome composition are numerous, including but not limited to nanoparticle tracking analysis, dynamic light scattering, resistive pulse sensing, atomic force microscopy, transmission electron microscopy, and flow cytometry, the studies are still not enough to explain the mechanism behind the effects and the applications in TCM are still limited. It is urgently necessary to carry out more comprehensive research and systematic characterization methods to understand exosomes in-depth, and broaden their applications. Due to the complexity of TCM and exosomes, there is still a long way to go.

Several problems need attention. First, after taking full account of cell sources that are used for collecting exosomes, strategies can be chosen to trigger them to secrete enough exosomes to ensure the downstream experiments. In the past few years, production has been found to be unsatisfactory. Furthermore, besides quantity, maintaining consistency in the quality of isolated exosomes, such as structural and biological integrity, is also an important thing. In addition, the separation process costs a lot. We expect that great progress will be shown in methods to harvest high-quantity and high-quality exosomes at low cost in the near future. Second, the purity of exosomes should be carefully verified, especially those exosomes secreted by TCM; otherwise, we could not distinguish specific differences between the drugs, cell debris, and exosomes and remove the risk of unwanted cargo. To improve the discrimination of purity, we need to clarify the classification of exosomes, thus providing a specific theoretical basis. Third, the current storage technique of exosome-based DDSs uses 4 °C for short-term storage and −80 °C for long-term storage, which causes unavoidable damage to the concentration, content, and integrity, because of the generation of ice crystals. Additionally, repeated freeze-thaw cycles lead to a decrease in proteins and an increase in particle size. Lastly, when conferred to clinical use, it is difficult to ensure the administration and safety of exosome-based DDSs. There are challenges in using current strategies to trace nano-sized materials precisely. Despite building a local or distant microenvironment in the body for enhancing function, the side effects of exosome-based DDSs could not be ignored when applied systemically. It is possible for DDSs to reach undesired regions. 

In summary, whether a monomer or a compound of TCM, exomes all belong to the treasure of the world when we use them properly. To apply the full potential of TCM, we ought to learn TCM from modern perspectives including not only exosome-based novel delivery systems, but also other advanced science.

## Figures and Tables

**Figure 1 molecules-27-07789-f001:**
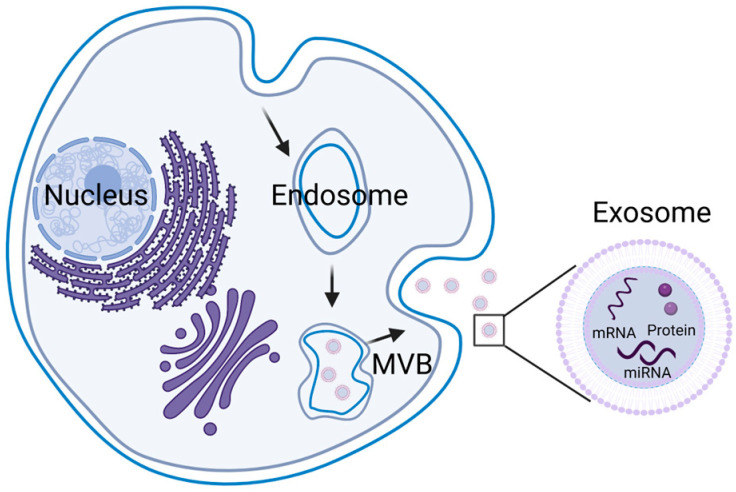
The origin and constituent of exosomes. Inside the cell, exosomes originated from the formed multivesicular bodies (MVBs) from endosomes. Their constituent comprises phospholipids, mRNA, miRNA, and proteins.

**Figure 2 molecules-27-07789-f002:**
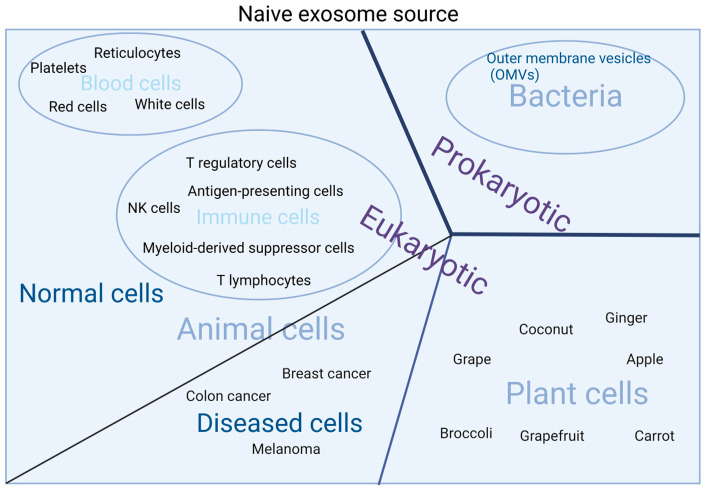
Primary sources of native exosomes.

**Figure 3 molecules-27-07789-f003:**
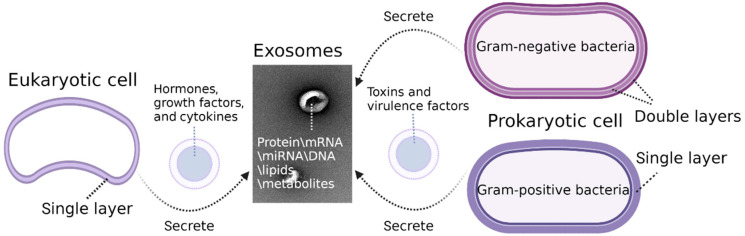
The similarities and differences between eukaryotic and prokaryotic exosomes.

**Figure 4 molecules-27-07789-f004:**
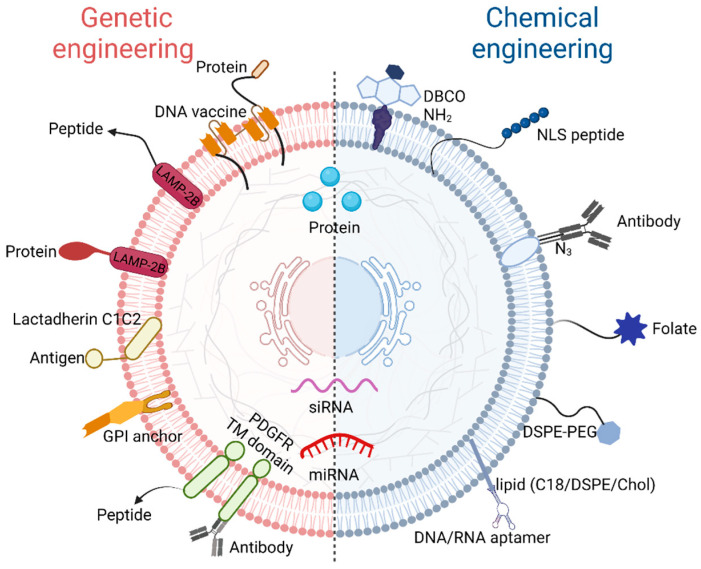
Modify exosomes via genetic and chemical engineering. Genetic engineering introduces proteins and peptides through the genetic fusion of membrane proteins. Chemical engineering introduces lipids, peptides, proteins, small molecules, and polymers through the chemical reactions between lipids and proteins or lipids and lipids.

**Figure 5 molecules-27-07789-f005:**
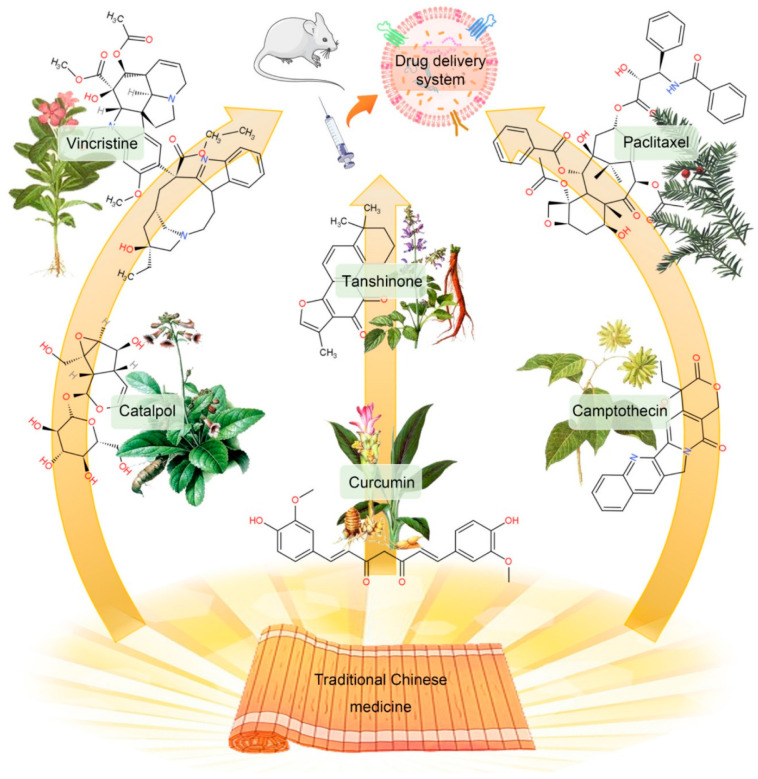
The monomeric active ingredients in TCM. Reproduced with permission from [124]. Copyright 2022, IVYSPRING.

**Table 1 molecules-27-07789-t001:** Exosome isolation methods.

Methods	Advantages	Disadvantages	Ref.
Differential ultracentrifugation and density gradient ultracentrifugation	Easy operation, and high purity	Dependence on expensive instruments, low automation, limited production, time consuming	[85,86,87,88,89]
Ultrafiltration	Simple and efficient without affecting the biological activity of exosomes	Uneconomical, ultrafiltration tubes required	[90,91]
Size exclusion chromatography	Good homogeneity of exosomes, less time consuming, more convenient, and more automatic	Need special equipment, not widely used	[92]
Immunoaffinity capture	High specificity, simple operation, and does not affect the integrity of exosome morphology	The biological activity is easily affected by pH and salt concentration	[93,94,95]
Membrane-based separation	High yield, and high speed	Impurity	[96,97]
Polyethylene glycol precipitation	Easy operation	More impurity proteins, non-uniform particle size	[98]
Microfluidic approach	Small consumption of samples and reagents, and a significant reduction in assay time	High cost	[99,100,101]

**Table 2 molecules-27-07789-t002:** Drug-loading methods of exosomes.

Loading Methods	Loading Drugs	Exosomes Source	Ref.
Incubation	PTX	Prostate cancer cell	[106]
	PTX and oncolytic adenovirus	Human lung cancer cell	[107]
	DOX	Human CRC cell	[108]
	DOX	BMDCs	[109]
Electroporation	DOX	Immature DCs	[66]
	SPIONs	B16-F10 melanoma cells	[110]
	Dextran	HeLa cervical cancer cells	[111]
Sonication	DOX	Raw 264.7 macrophages	[112]
	PTX/DOX	Raw 264.7 macrophages	[113]
	Small RNAs	HEK293T and MCF-7	[114]
Extrusion	Tegafur	Attenuated *Salmonella*	[115]
	MOF-protein	MDA-MB-231 tumor cells	[116]
Freeze-thaw	ICG and R837	CD47-overexpressed CT26 cells	[117]
Acoustofluidics	DOX	Human plasma	[118]
Hypotonic dialysis	DOX Nucleic acid	BM-MSCs HEK293T	[119] [120]

Abbreviations: DOX: doxorubicin; PTX: paclitaxel; CRC: colorectal cancer; BMDCs: bone marrow dendritic cells; DCs: dendritic cells; SPIONs: superparamagnetic iron oxide nanoparticles; MOF: metal−organic frameworks; ICG: indocyanine green; BM-MSCs: Bone marrow mesenchymal stromal cells.

## Data Availability

No new data were created or analyzed in this study. Data sharing is not applicable to this article.

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
