# Peer review of "Exosomes as Novel Delivery Systems for Application in Traditional Chinese Medicine"

_molecules, 2022, doi:10.3390/molecules27227789_

Round 1

Reviewer 1 Report

The authors presented a review focusing on the use of exosomes in traditional Chinese medicine as drug delivery systems. In its present form the manuscript contains information on the classification, isolation, drug loading mechanisms and use in TCM of the exosome but lacks of a clear discussion. Authors should focus on the review to reach a more sophisticated level without underestimating the importance of their vision of expertise, especially considering the traditional Chinese Medicine, to support their claim. Some points can be mentioned for a potential progress,

1.     Authors should consider revising the language of the manuscript to reach a higher level. Adding connecting sentences between sections will support the flow of the MS.

2.     Authors should review the reference notation at the end of the sentences.

3.     In the abstract, the second sentence (page 1, line 11-13) should be more descriptive.

4.     Figure 1 caption should can be more descriptive to catch the attention of the reader.

5.     Page 2, line 65-66, it has been said that exosomes are secreted only from blood, immune and cancer cells. However, exosomes are released from various cells such as fibroblasts, immune system cells (T- cells, B-cells, dendritic cells), adipocytes, and tumor cells and isolated from all normal and diseased body fluids such as blood, urine, breast milk, amniotic fluids, and bronchial alveolar lavage. Authors should revise this information.

6.     Page 3, line 109 “A summary of the naive exosome source is given in Figure 2” should be moved to end of the first paragraph of section 2.1 (page 2, line 64).

7.     First paragraph of section 2.2 (page 4, line 113-114) is incomprehensible; it needs to be strengthen and enhanced.

8.     In section 3.1 only isolation methods and their advantages and disadvantages are mentioned. However, the necessity of choosing which method in which case is not mentioned. Also, the usage of references is insufficient.

9.     In table 1, chromatography-based and membrane-based techniques are not mentioned.

1.  In section 3.2, advantages and disadvantages of drug loading methods are not mentioned. Also, it should be explained which technique is endogenous and which is exogenous.

11.  In table 2, other loading methods (transfection, surface treatment, hypotonic dialysis etc.) are not included.

1.  In section 4, after specifying TCM as the abbreviation of Traditional Chinese medicine, it is necessary to write it as TCM in the following paragraphs.

1.  In section 4.1, PTX as an abbreviation for Paclitaxel should be mentioned on line 322.

1.  Figure 5 (Page 10, line 330) is not included in the manuscript.

Author Response

Point 1: The authors presented a review focusing on the use of exosomes in traditional Chinese medicine as drug delivery systems. In its present form the manuscript contains information on the classification, isolation, drug loading mechanisms and use in TCM of the exosome but lacks of a clear discussion. Authors should focus on the review to reach a more sophisticated level without underestimating the importance of their vision of expertise, especially considering the traditional Chinese Medicine, to support their claim. Some points can be mentioned for a potential progress,

Response 1: Thanks for the comment. We agree with the review’s points. In our original manuscript, we listed and summarized the research of others in Sections 1-4 and discussed mainly in Section 5. To make progress, we have tried our best to twist our own opinions with the evaluation of publications together. To support claims, we also turn to some experts in TCM for help. Relevant revisions have been marked using the “Track Changes” function of MS word.

Point 2: Authors should consider revising the language of the manuscript to reach a higher level. Adding connecting sentences between sections will support the flow of the MS.

Response 2: We apologize for the poor language of our manuscript. We moved sentences and sections which lead to poor readability. We have now worked on both language and readability and have also involved native English speakers for language corrections. We hope that the flow and language level have been substantially improved.

Point 3: Authors should review the reference notation at the end of the sentences.

Response 3: Thanks for the reminder. We have checked the reference notation at the end of the sentences carefully.

Point 4: In the abstract, the second sentence (page 1, line 11-13) should be more descriptive.

Response 4: We have replaced the original sentence “On account of the structure containing constituents of the parental cells that secrete them and the ability to participate in the regulation of cellular communication, exosomes are taken into account as drug delivery systems (DDSs).” into “ They are composed of a lipid membrane bilayer structure containing different constituents, such as the surface ligands and receptors, from the parental cells. Depending on the variety of sources, exosomes have the ability to participate in a diverse range of biological processes, including the regulation of cellular communication. On account of their ideal native structure and characteristics, exosomes are taken into account as drug delivery systems (DDSs).”

Point 5: The figure 1 caption should be more descriptive to catch the attention of the reader.

Response 5: We have added more description. Now, the figure 1 caption is “The origin and constituent of exosomes. Inside the cell, exosomes originated from the formed multivesicular bodies (MVBs) from endosomes. Their constituent comprises phospholipids, mRNA, miRNA, and proteins.”

Point 6: Page 2, line 65-66, it has been said that exosomes are secreted only from blood, immune and cancer cells. However, exosomes are released from various cells such as fibroblasts, immune system cells (T- cells, B-cells, dendritic cells), adipocytes, and tumor cells and isolated from all normal and diseased body fluids such as blood, urine, breast milk, amniotic fluids, and bronchial alveolar lavage. Authors should revise this information.

Response 6: We are sorry for making the reviewer confused about our description. We want to express that, exosomes from blood, immune, and cancer cells, are representatives of animal exosomes. We have revised this description.

Point 7: Page 3, line 109 “A summary of the naive exosome source is given in Figure 2” should be moved to end of the first paragraph of section 2.1 (page 2, line 64).

Response 7: Thanks for the suggestion. We have moved the sentence to the end of the first paragraph of section 2.1.

Point 8: First paragraph of section 2.2 (page 4, line 113-114) is incomprehensible; it needs to be strengthen and enhanced.

Response 8: We have changed the original sentence “To make exosomes with more of the desirable features, genetic and chemical engineering have been developed to modify exosomes (Figure 3).” with “ Irrespective of the parent cell, exosomes have some common features, including the expression of certain tetraspanins (CD9, CD63, and CD81), heat shock proteins (Hsp 60, Hsp 70, and Hsp 90), biogenesis-related proteins (Alix and TSG 101) and so on [Cell Biosci 9, 19 (2019)]. We can also check the collected data in Vesiclepedia (http://www.microvesicles.org), Exocarta (http://www.exocarta.org), and a Bioinformatics lab from China (http://bioinfo.life.hust.edu.cn), to know their distinguishing characteristics [Cell Commun Signal 20, 145 (2022)]. Although there are thousands of characteristics that exosomes could provide by nature, their practical applications are still limited in some cases. Therefore, to make exosomes with more of the desirable features, genetic and chemical engineering have been developed to modify exosomes (Figure 4).”

Point 9: In section 3.1 only isolation methods and their advantages and disadvantages are mentioned. However, the necessity of choosing which method in which case is not mentioned. Also, the usage of references is insufficient.

Response 9: We have referred to more references and added the discussion “ From what have discussed above, the choice of which isolation method in which case should depend on the downstream application together with the amounts of starting materials. If the application needs a high quantity of exosomes, the precipitation method is the best. If the size range matters, ultracentrifugation, size exclusion, chromatography, and immunoaffinity capture can be chosen for yielding a narrow size range for exosomes. If starting materials are limited, methods that cause small consumption of samples need to be picked, such as ultrafiltration and microfluidic approach.” into section 3.1.

Point 10: In table 1, chromatography-based and membrane-based techniques are not mentioned.

Response 10: We have listed size exclusion chromatography and immunoaffinity capture in table 1. The former is a chromatography-based technique while the latter is a membrane-based technique.

Point 11: In section 3.2, advantages and disadvantages of drug loading methods are not mentioned. Also, it should be explained which technique is endogenous and which is exogenous.

Response 11: Thanks for this useful suggestion. We have perfected section 3.2 according to the suggestion.

Point 12: In table 2, other loading methods (transfection, surface treatment, hypotonic dialysis etc.) are not included.

Response 12: We listed in the article the following loading methods, incubation, electroporation, sonication, extrusion, freeze-thaw, and acoustofluidics. Transfection could be accomplished by incubation or electroporation. Therefore, we didn’t mention it in table 2. We included surface treatment, which is related to modified exosomes, in Section 2.2. We accepted the reviewer’s good suggestion and added hypotonic dialysis in table 2. Besides, we have added discussions about hypotonic dialysis in our manuscript “Hypotonic dialysis relies on a transmembrane pH gradient to encapsulate weakly basic or negatively charged small-molecule drugs effectively. Lin J. et al. prepared exo-some-loaded doxorubicin (Exo-Dox) by mixing both of them, desalinizing them with tri-ethylamine, and dialyzing them with PBS overnight [Int J Nanomedicine. 14, 8603-8610 (2019)]. Jay S. M. et al. exposed exosomes to dehydration by ethanol for miRNA loading. Some high molecular weight protein content in exosomes may degrade using the dialysis steps [Mol Ther. 28, 3, 975-985 (2020)].”.

Point 13: In section 4, after specifying TCM as the abbreviation of Traditional Chinese medicine, it is necessary to write it as TCM in the following paragraphs.

Response 13: We have carefully checked the full text and replaced all the following statements except the first one of Traditional Chinese medicine into TCM.

Point 14: In section 4.1, PTX as an abbreviation for Paclitaxel should be mentioned on line 322.

Response 14: Thanks for pointing out the omission. We have mentioned it in the text.

Point 15: Figure 5 (Page 10, line 330) is not included in the manuscript.

Response 15: We are sorry for the omission. We have corrected this error.

Reviewer 2 Report

In this review, the authors discuss the classification and application of exosomes as drug delivery systems in traditional Chinese medicine. This is a comprehensive well organized review with a pleasant flow. Although the authors provide a large overview of the different approaches used for construction of exosomes as drug delivery systems and drug-loading methods, this review requires further refinement in the content of the information presented. The scope of this review is clear but needs to be more supported by scientific data.

Comments by Section

Abstract  is a good overview of the topic

Section 1

1. The information described in this section was appropriate and exhaustive to introduce the following sections.

Section 2

2. It is interesting the distinction of eukaryotic and prokaryotic exosomes and their primary sources. However, what is the biological composition of naive exosomes in the inner cavity? Are there difference in the inner content between eukaryotic and prokaryotic exosome. Does this difference impact on the final application of exosomes as drug delivery systems? Maybe, a table or a schematic might be useful in this respect.

3. Current methods to characterize exosomes and exosomes composition would also be a nice addition to the review.

4. Hybrid exosomes with natural lipids have been developed as you reported in raw 150-151. Which natural lipids help to enhance the accumulation of this DDS at the desired location? Are there any other to avoid? A better section on this might strengthen your review.

Section 3

5. Regarding the section on exsosomes as drug delivery systems, what are the percentage of uptake of these SSDs? Are there exosomes that are better uptaken rather than others?

6. What is the yield of the drug loaded in relation of the different drug loading methods?

Section 4

7. In the application section, how is the stability of the exosomes as drug delivery system over time and upon temperature? Does the inner content remain functional? How long do they remain active as drug delivery systems?

8. Does the release of the payload change depending on the cell source? Which methods are usually employed to release the payloads?

Author Response

Point 1: In this review, the authors discuss the classification and application of exosomes as drug delivery systems in traditional Chinese medicine. This is a comprehensive well organized review with a pleasant flow. Although the authors provide a large overview of the different approaches used for construction of exosomes as drug delivery systems and drug-loading methods, this review requires further refinement in the content of the information presented. The scope of this review is clear but needs to be more supported by scientific data.

Response 1: Thanks for reviewer’s recommendation. We have refined the content and added more references in order to introduce more scientific data.

Point 2: Abstract  is a good overview of the topic. The information described in this section 1 was appropriate and exhaustive to introduce the following sections.

Response 2: We thank the reviewer’s affirmation of our work.

Point 3: Section 2-It is interesting the distinction of eukaryotic and prokaryotic exosomes and their primary sources. However, what is the biological composition of naive exosomes in the inner cavity? Are there difference in the inner content between eukaryotic and prokaryotic exosome. Does this difference impact on the final application of exosomes as drug delivery systems? Maybe, a table or a schematic might be useful in this respect.

Response 3: This is a good question. Protein, mRNA, miRNA, DNA, lipids, and metabolites are the biological composition of naive exosomes in the inner cavity [Drug Deliv. and Transl. Res. 12, 1047–1079 (2022)]. They vary depending on the source and the original isolation or enrichment techniques. Eukaryotic exosomes are secreted by cells that are surrounded by a single membrane. Typically, they have soluble factors in the inner content such as hormones, growth factors, and cytokines that can act on the source cell itself. Prokaryotic exosomes, focused on the pathogenic bacteria, are secreted by Gram-negative bacteria that possess two phospholipid membranes and Gram-positive bacteria that are surrounded only by a single membrane, which contains toxins and virulence factors, such as the lipopolysaccharides (LPS) [J. Extracell. Vesicles 4, 27066 (2015)]. The difference indeed has impacts on the final application of exosomes as drug delivery systems. We accepted the reviewer’s suggestion and added a schematic to the Figure 3.

Point 4: Current methods to characterize exosomes and exosomes composition would also be a nice addition to the review.

Response 4: We agree with the reviewer's view. The current methods to characterize exosomes and exosome composition are a lot, including but not limited to nanoparticle tracking analysis, dynamic light scattering, resistive pulse sensing, atomic force microscopy, transmission electron microscopy, and flow cytometry. Unfortunately, because we want to focus on the last section, the previous sections are limited in length. We have mentioned current methods briefly in the text.

Point 5: Hybrid exosomes with natural lipids have been developed as you reported in raw 150-151. Which natural lipids help to enhance the accumulation of this DDS at the desired location? Are there any other to avoid? A better section on this might strengthen your review.

Response 5: This DDS used platelet membrane to modify stem cell exosome. Platelet membranes are lipid bilayers having membrane integrin receptors, such as GPVI, participating in the natural “injury finding”. Hence, platelet membrane-hybrid exosomes can target vascular injury under myocardial infarction. As compared to non-modified exosomes, hybrid exosomes notably enhanced the cellular uptake in endothelial cells and cardiomyocytes, but not in macrophages. Furthermore, platelet membranes improve the therapeutic capacity of stem cell exosomes.

Liver accumulation is undesirable and needs to avoid for this DDS that is not intended for liver diseases. But Cheng K. et al. found a steady concentration of platelet membrane-hybrid exosomes in the liver. Their future studies will try to finetune the exosomes to reduce liver off-target loss.

We worked on the manuscript for adding the above discussion. We hope that this review has been substantially improved.

Point 6: Section 3-Regarding the section on exsosomes as drug delivery systems, what are the percentage of uptake of these DDSs? Are there exosomes that are better uptaken rather than others?

Response 6: Many scientists have been interested in the development of exosome-based DDSs for improving the uptake. For example, Piffoux et al. pointed liposome–exosome hybrid DDS structure could improve cellular uptake of anti-cancer photosensitizer in CT26 colon cancer cells from approximately 20% to 70% [ACS Nano 12, 6830−6842 (2018)]. Lang et al. demonstrated cardiomyocytes exhibited increased uptake of cardiomyocyte specific peptide-modified exosomes by two-fold when compared with non-targeted exosomes [Sci. Rep. 9, 10041 (2019)]. Likewise, Ou et al. investigated cellular uptake mechanisms of exosomes showing an 8-to-40-fold higher cellular internalization than liposomes, mainly through receptor-mediated processes (i.e., clathrin- and caveolae-mediated endocytosis) [Pharmaceutics 14, 1738 (2022)]. We have added this discussion in the introduction of Section 3.

Point 7: What is the yield of the drug loaded in relation of the different drug loading methods?

Response 7: Thanks for the question. We have added the loading efficiency of different loading methods in Section 3.2.

Point 8: Section4-In the application section, how is the stability of the exosomes as drug delivery system over time and upon temperature? Does the inner content remain functional? How long do they remain active as drug delivery systems?

Response 8: Evidence to date suggests that the stability of the exosomes as drug delivery systems over time and upon temperature is good [AAPS J. 20, 1 (2018)]. They extend the drug’s half-life and increase drug release stability by encapsulating the drug inside the lipid bilayer membrane structure. It was reported, under freezing conditions (–20 °C and –80 °C), no significant change happened in the inner content [J. Extracell. Vesicles 7, 1535750 (2018)]. For up to two years, exosomes remain active as drug delivery systems [Drug Deliv. 28,1501-1509 (2021)]. However, challenges with respect to the stability and storage of the exosome-based DDSs still persist because this mode may be limited by cost.

Point 9: Does the release of the payload change depending on the cell source? Which methods are usually employed to release the payloads?

Response 9: Yes, it does. The diversity of exosomal-payload and their functions may provide future pioneering target treatments. For one method, we can adjust pH and salt concentration to trigger the release of payloads [Biotechnol. Adv. 31, 543-51 (2013)]. For another, if exosomes are modified with stimuli-responsive ligands, we can use various types of stimuli to release the payloads including force, light, polarity, temperature, electricity, ion, pH, etc [WIREs Nanomed. Nanobiotechnol. 9, e1450 (2017)].

Reviewer 3 Report

The manuscript is well written and introduces the topic of new research but lacks in several areas; these issues must be addressed to make it interesting for the journal readers.

·       Lacks graphical abstract in the manuscript.

·       There are many reviews written on Exosomes as novel delivery systems; There is a need to write some points that differentiate from other reviews.

·       The review is not revealed the search strategies, inclusion and exclusion criteria and risk of bias assessment for individual studies therefore, there is a need to add a material and methods section. What is the timeline of the review?

·       Summarizes the most recent laboratory and clinical findings on exosomes across numerous medical disciplines, thereby offering readers a broad-ranging and solid foundation for prospective investigative efforts.

·       Make a new section to describe the basic mechanisms of Exosomes.

·       New section 4.4 has also been required to elaborate on the function and therapeutic use of exosomes in bacterial and viral infections in the context of TCM.

·       Add the conclusive remarks together with future recommendations in the separate section in Conclusion.

Author Response

Point 1: The manuscript is well written and introduces the topic of new research but lacks in several areas; these issues must be addressed to make it interesting for the journal readers.

Response 1: Thanks for the reviewer’s suggestion. We accepted the suggestion and corrected the manuscript carefully, all the changes/additions to the manuscript are given using the “Track Changes” function of MS word.

Point 2: Lacks graphical abstract in the manuscript.

Response 2: We have added a graphical abstract in the manuscript

Point 3: There are many reviews written on Exosomes as novel delivery systems; There is a need to write some points that differentiate from other reviews.

Response 3: We agree with the reviewer's comment that exosomes as novel delivery systems is a very popular topic and special points that differ from other reviews should be shown in our review. Hence, we prepared this review in an organized mode, referring to the classification of exosomes, the construction of exosome-based DDSs, and the application of exosome-based DDSs for the treatment of different diseases in TCM. TCM has been commonly used in Asia for thousands of years. But fewer studies focus on exosome-based DDSs to deliver natural compounds when compared to western medicine. This review summarized relevant literature reports in order to enlighten researchers to better understand the link between exosomes and TCM and to contribute more meaningful work.

Point 4: The review is not revealed the search strategies, inclusion and exclusion criteria and risk of bias assessment for individual studies therefore, there is a need to add a material and methods section. What is the timeline of the review?

Response 4: As this review involves many research fields, we summarized nearly 20 years in recent years of relevant developments based on the best of our knowledge from various official websites, not using meta-analysis, because incomplete highly cited results conducted in PubMed or Google in a fixed range. Therefore, the materials and methods section mentioned by the reviewer is not appropriate in the text. In order to make up and make the review look more scientific, we mentioned related content about the search strategies in brief in Section 1. Besides, we also listed some databases about exosomes in the text for the convenience of readers, such as Vesiclepedia (http://www.microvesicles.org), Exocarta (http://www.exocarta.org), and a Bioinformatics lab from China (http://bioinfo.life.hust.edu.cn).

Point 5: Summarizes the most recent laboratory and clinical findings on exosomes across numerous medical disciplines, thereby offering readers a broad-ranging and solid foundation for prospective investigative efforts.

Response 5: Thanks for the insightful suggestion. We have tried our best to summarize the most recent laboratory and clinical findings on exosomes across numerous medical disciplines throughout this review.

Point 6: Make a new section to describe the basic mechanisms of Exosomes.

Response 6: It’s noted that in the introduction of Section 2, we discussed the basic mechanism of exosomes before the classification part. We also have added more details in this part.

Point 7: New section 4.4 has also been required to elaborate on the function and therapeutic use of exosomes in bacterial and viral infections in the context of TCM.

Response 7: We think the contents about bacterial and viral infections could be divided into section 4.3 which title is “others”. In section 4.3, we listed inflammatory arthritis and inflammatory bowel disease caused by bacteria, viruses, and other microorganisms invading as examples.

Point 8: Add the conclusive remarks together with future recommendations in the separate section in Conclusion.

Response 8: We accepted the reviewer’s suggestion, changed the final part “Summary and perspectives” into “Conclusion”, and added the conclusive remarks together with future recommendations in a separate section.

Round 2

Reviewer 1 Report

Compared to the previous revision, the authors made corrections and additions about the points and have a progress. For a better enhancement, some minor changes should also be considered.

·  * The sentence added in section 3.1 regarding the selection of isolation methods (line 290-295) should be based on reference and more explanatory information should be given.

·   * In Table 2 “Immunoaffinity capture isolation technique” is based on surface biomarkers and isolation is achieved by coating the magnetic beads with antibodies targeting proteins located on the exosome surface. However, membrane-based separation is a new isolation technology that is based on the properties of the exosome phospholipid bilayer membrane. Authors should consider giving more information about the new technique.

Author Response

Point 1: Compared to the previous revision, the authors made corrections and additions about the points and have a progress. For a better enhancement, some minor changes should also be considered.

Response 1: We thank the reviewer’s affirmation of our work. We’d like to improve our manuscript.

Point 2: The sentence added in section 3.1 regarding the selection of isolation methods (line 290-295) should be based on reference and more explanatory information should be given.

Response 2: Thanks for the suggestion. We have added references and more explanatory information in this part.

Point 3: In Table 1 “Immunoaffinity capture isolation technique” is based on surface biomarkers and isolation is achieved by coating the magnetic beads with antibodies targeting proteins located on the exosome surface. However, membrane-based separation is a new isolation technology that is based on the properties of the exosome phospholipid bilayer membrane. Authors should consider giving more information about the new technique.

Response 3: Thanks for the suggestion. We have updated Table 1 and graphical abstract, and added a related paragraph, “Membrane-based separation is a new and powerful isolation technology that is implemented by negatively charged phosphate groups in lipid bilayer structures of exosomes [Chem. Sci. 2019, 10: 1579–1588; Front. Bioeng. Biotechnol. 2022, 9: 811971]. Some metal oxides (e.g., TiO2) and positively charged molecules (e.g., protamine) can easily bind to the phosphate groups. Because exosomes are rich in phosphate groups, this strategy for isolating exosomes has obvious advantages in high yield and speed. However, for some similar structures of exosomes in the microenvironment which exosomes lies, membrane-based separation may result in other impurities.”, in the manuscript.

Reviewer 2 Report

Most of the suggestions have been incorporated by the authors in the revised manuscript. Therefore, no issue with considering it for publication.

Author Response

We thank the reviewer’s affirmation of our work.

Reviewer 3 Report

·       Most of the suggestions have been incorporated by the authors in the revised manuscript. Therefore, no issue with considering it for publication.

Author Response

(The authors gave the same response as above.)
